# Inhibition of Advanced Glycation End-Products by *Tamarindus indica* and *Mitragyna inermis* Extracts and Effects on Human Hepatocyte and Fibroblast Viability

**DOI:** 10.3390/molecules28010393

**Published:** 2023-01-02

**Authors:** Relwendé Justin Ouédraogo, Umair Aleem, Lassina Ouattara, Muhammad Nadeem-ul-Haque, Georges Anicet Ouédraogo, Humera Jahan, Farzana Shaheen

**Affiliations:** 1Unit of Training and Research in Life and Earth Sciences, Department of Biochemistry-Microbiology, Nazi Boni University, Bobo-Dioulasso 01 BP 1091, Burkina Faso; 2Laboratory of Research and Teaching in Animal Health and Biotechnology, Nazi Boni University, Bobo-Dioulasso 01 BP 1091, Burkina Faso; 3Dr. Panjwani Center for Molecular Medicine and Drug Research, International Center for Chemical and Biological Sciences, University of Karachi, Karachi 75270, Pakistan; 4Third World Center for Science and Technology, Hussain Ebrahim Jamal Research Institute of Chemistry, University of Karachi, Karachi 75270, Pakistan

**Keywords:** *Mitragyna inermis*, *Tamarindus indica*, sweroside, antiglycation, cytotoxicity

## Abstract

*Tamarindus indica* and *Mitragyna inermis* are widely used by herbalists to cure diabetes mellitus. The aim of this study is to investigate the inhibitory potential of aqueous and various organic solvent fractions from both plants and some isolated compounds against advanced glycation end-products (AGEs). For this purpose, an in vitro BSA–fructose glycation model was used to evaluate the inhibition of AGE formation. Furthermore, the effects of the fractions on mouse fibroblast (NIH-3T3) and human hepatocyte (HepG2) survival were evaluated. The leaf, stem, and root fractions of both plants exhibited significant inhibition of AGEs formation. The IC_50_ values appeared to be less than 250 µg/mL; however, all fractions presented no adverse effects on NIH-3T3 up to 500 µg/mL. Otherwise, our phytochemical investigation afforded the isolation of a secoiridoid from the *Mitragyna* genus named secoiridoid glucoside sweroside (**1**), along with three known quinovic acid glycosides: quinovic acid-3*β*-*O*-*β*-d-glucopyranoside (**2**), quinovic acid-3-*O*-*β*-d-6-deoxy-glucopyranoside, 28-*O*-*β*-d-glucopyranosyl ester (**3**), and quinovic acid 3-*O*-*α*-l-rhamnopyranosyl-(4→1)-*β*-d-glucopyranoside (**4**). In particular, **1**–**3** are compounds which have not previously been described in *Mitragyna inermis* roots. However, the isolated compounds did not exhibit AGE inhibitory activity. Further investigation on these potent antiglycation fractions may allow for the isolation of new antidiabetic drug candidates.

## 1. Introduction

Diabetes mellitus (DM) is a multi-factorial disease, characterized by uncontrolled blood glucose levels due to shortcomings in insulin production and/or its function. It has become a serious health challenge worldwide. According to the World Health Organization (WHO), diabetes mellitus incidence is increasing so fast that the number may reach up to 700 million by 2045 without appropriate treatment. The high prevalence of diabetes is reported to be associated with obesity, unhealthy diets, and sedentary lifestyles [1]. According to several studies, elevated blood sugar levels—also known as hyperglycemia—play a crucial role in the causation of several diabetes-associated late complications, as well as the excessive formation of advanced glycation end-products (AGEs) [2].

AGEs are the final products of the Maillard reaction, a non-enzymatic reaction which takes place between free amino groups of proteins and carbonyl groups of reducing sugars (e.g., glucose, fructose, and ribose) to produce Schiff bases. The subsequent rearrangement of Schiff bases leads to the formation of more stable Amadori products, which further undergo a series of transformations (e.g., oxidation, condensation, dehydration, cyclization, and so on) to irreversibly form a heterogenous group of pathogenic adducts, known as advanced glycation end-products (AGEs). It has been reported that the excessive AGE formation in the hyperglycemic environment may cause alterations in structural and functional properties of numerous proteins, such as albumin, collagen, elastin, and tubulin, among others [3,4,5]. In addition, AGEs are considered an important contributing factor in the initiation and progression of the inflammatory response, through their interaction with their receptor, RAGE, expressed on the surface of various cell types, including macrophages [1,5]. Moreover, AGEs have been reported to be involved in various health disorders, including diabetes, renal failure, and neurodegenerative diseases [2,6]. Therefore, targeting the formation of AGEs by identifying new and potential antiglycation products with low cytotoxicity may lead to significant success in achieving this aim.

*Tamarindus indica* Linn. (Caesalpiniaceae) is a pan-tropical species widely distributed throughout the tropical belt, from Africa to South Asia, northern Australia, and throughout Southeast Asia, Taiwan, and China [7]. In Burkina Faso, this species has been observed in three phytogeographic spaces; namely, the Sudanese north, the sub-Sahel, and the Sudanese south [8].

From Senegal to Cameroon, Sudan, Benin, Togo, and the Democratic Republic of Congo, *Mitragyna inermis* (Willd) O. Kuntze (Rubiaceae) is sometimes found in pure settlements. It is found everywhere in Burkina Faso where there are still water reservoirs. The plant is a small bushy tree or shrub, high, and often very branchy from the base [7].

Ethnobotanical surveys have revealed the uses of *M. inermis* and *T. indica* in the treatment of several diseases, including metabolic illnesses such as diabetes and hypertension [9,10,11]. Moreover, stem, bark, and leaf extracts of these plants have been reported to exhibit antidiabetic effects in vivo, as well as relevant pro-diabetes enzyme inhibition [12,13,14]. To the best of our knowledge, there has been no reported research on the ability of products derived from these plants in terms of managing diabetes complications. However, phytochemical reports have shown that these plants contain phenolic compounds, flavonoids, terpenoids, alkaloids, and so on [15,16,17,18]. Several studies have reported that natural products, such as flavonoids and flavonoid-rich fractions, which possess potent antioxidation activity, are able to prevent the formation of AGEs in vitro [19,20,21,22,23,24]. Phenolic compounds, due to their antioxidant properties, have presented advantages in antiglycation drug research. In this line, our previous study detailed the sampling of both plants, showing a high correlation between phenolic compounds content and antioxidant potential [25].

In this report, we investigated the effects of *M. inermis* and *T. indica* extract fractions in a fructose–BSA glycation model and their cytotoxicity status. Furthermore, a previously undescribed compound from *M. inermis* (**1**), as well as some known compounds (**2**–**4**) without any reported effect on diabetes complications model, were also investigated in the fructose–BSA glycation model.

## 2. Results and Discussion

### 2.1. Antiglycation Activity of Mitragyna inermis and Tamarindus indica Fractions

For the present study, different fractions, denoted **Frac1** (decoction of *M. inermis* leaves), **Frac2** (additional ethyl acetate + butanol + acetone fractions of *M. inermis* leaves), **Frac3** (additional ethyl acetate + acetone fractions of *M. inermis* stem), **Frac4** (additional ethyl acetate + acetone fractions of *M. inermis* root), **Frac5** (additional ethyl acetate + butanol fractions of *T. indica* leaves), **Frac6** (acetone fraction of *T. indica* leaves), **Frac7** (additional ethyl acetate + acetone fractions of *T. indica* stem), and **Frac8** (additional ethyl acetate + acetone fractions of *T. indica* root), were obtained from the leaves, stem, and roots of *M. inermis*, and *T. indica*, and were tested against a fructose-mediated BSA glycation model. The concentrations of fractions tested for the antiglycation activity were either 250, 500, or 1000 µg/mL. Our results indicated that the degree of antiglycation activity of the tested fractions varied considerably among the different parts (i.e., stem, leaves, and roots) of *M. inermis* and *T. indica* (Table 1). Among them, the highest degree of antiglycation activity was measured for **Frac7** (91.82 ± 0.21% inhibition) at 1000 µg/mL (*p <* 0.0001). Other fractions of *T. indica* also showed a significant inhibition effect against the fructose-mediated BSA glycation model. In contrast, **Frac3** and **Frac4**, derived from the stem and roots of *M. inermis*, respectively, exhibited a slightly lower inhibition (68.04 ± 0.78% inhibition, and 68.34 ± 0.47% inhibition at 1000 µg/mL, respectively), as compared to **Frac1** and **Frac2** derived from *M. inermis* leaves, as detailed in Table 1. The variation in antiglycation activity among these derived fractions from the different parts (i.e., stem, leaves, and roots) of the respective plants may be due to the differences in available active constituents. Therefore, the exhibited activity of the derived leaf fractions of *M. inermis* might be due to the presence of relatively abundant active compounds, compared to the fractions derived from roots. Moreover, all fractions showed a concentration-dependent effect on AGE inhibition, with their IC_50_ found to be less than 250 µg/ mL. However, flavonoid compounds are known to have both antioxidant and antiglycation activities [23,24]. The screening showed different classes of flavonoids in all fractions. This could justify the activities shared by the fractions. Furthermore, that statement is in agreement with the antioxidant and flavonoid contents reported for *M. inermis* and *T. indica* [25,26,27,28,29]. These data suggest that further studies are required to understand the abilities of such tractions to prevent glycation in vivo and complications related to diabetes such as atherosclerosis, cardiac dysfunction, and vascular inflammation.

### 2.2. Cytotoxicity Profile of Mitragyna inermis and Tamarindus indica Fractions in HepG2 and NIH-3T3 Cells

*M. inermis* and *T. indica* fractions were evaluated for their cytotoxicity in human hepatocellular carcinoma (HepG2) cells at the concentration range from 15–1000 µg/mL by employing an MTT-based colorimetric assay. The cellular toxicity of these fractions was also evaluated against a normal fibroblast (NIH-3T3) cell line. It is well known that the liver is involved in the detoxification and metabolism of drugs through the cytochrome P450 (CYP450) pathway. Hence, the withdrawal of various drugs from clinical trial studies is often due to their adverse effects in liver cells [30]. Therefore, it was necessary to determine the toxicological effects of the considered fractions on various cell lines. Furthermore, the use of in vitro cellular-based models for toxicological studies has been shown to be a useful and economical approach, as compared to studies based on animal models [30]. The results demonstrated that the degree of the different fractions derived from the respective plants’ cellular toxicity varied significantly from cell line to cell line. In NIH-3T3 cells, **Frac1**, **Frac4**, and **Frac5** (with 38.04 ± 3.26%, 25.83 ± 0.30%, and 46.66 ± 0.94% inhibitions, respectively) were found to be non-toxic to a concentration of 500 µg/mL, while **Frac3** (with 42.92 ± 1.44% inhibition) was found non-cytotoxic up to a concentration of 1000 µg/mL. In contrast, **Frac2**, **Frac6**, **Frac7**, and **Frac8** were found to present a toxic effect at concentrations above 125 µg/mL, as shown in Figure 1. On the other hand, in HepG2 cells, **Frac2**, **Frac3**, **Frac6**, and **Frac7** (with 32.86 ± 3.88%, 22.16 ± 3.76%, 41.54 ± 3.06%, and 41.71 ± 1.82% inhibitions, respectively) were found to be non-toxic up to a concentration of 500 µg/mL, while the fraction **Frac4** (with 37.50 ± 1.75% inhibition) was found to be non-cytotoxic up to a concentration of 1000 µg/mL. In contrast, fractions **Frac1**, **Frac5**, and **Frac8** were found to present a toxic effect on HepG2 at concentrations above 250 µg/mL, as shown in Figure 2. However, the toxicity exhibited by some fractions at particular concentrations against both HepG2 and 3T3 cells might be due to the use of higher concentrations on the mono-layer culture of cells. The low cytotoxicity and transforming potential in this study are in agreement with previous findings [15,16,17,18,31,32].

### 2.3. Compounds Isolated from M. inermis Fraction and Their Anti-Glycation Activities

Column chromatography of **Frac4**, which presented potent antiglycation ability, afforded four compounds. Compound **1** is sweroside, while compounds **2**–**4** are triterpenoids.

Compound **1** (secoiridoid glucoside sweroside) was isolated as a yellow amorphous powder with molecular weight 358.1511 and molecular formula C_16_H_21_O_9_. The peak at *m/z* = 195 revealed the presence of Secoiridoid (C_10_H_10_O_4_), seemingly with the loss of one monosaccharide *m/z* = 163 (C_6_H_11_O_5_). HSQC spectra confirmed the presence of one monosaccharide in that compound. The signal at *δ* = 7.59 (d, *J* = 2.5 Hz) is typical to H-3 of secoiridoid aglycone [33]. The HMBC correlation of Glc H-1′ *δ* = 4.67 (d, *J* = 7.9 Hz) with the aglycone C-1 *δ* = 98.0 established the monosaccharide C-1′ at *β* position to C-1 of the aglycone. ^13^C data showed 16 C, 4 CH_2_, 2 quaternaries. The FT-IR data confirmed the presence of O–H (3326.30 cm^−1^) and C–H (2943.41–2831.56 cm^−1^) in that compound. It presented maximum absorption at 243 nm, typical of that found and reported previously [33]. NOESY spectrum analysis revealed a correlation between H-1, H-8, and H-10, while a correlation was also seen between H-6 and H-3′, H-3′ and H-1′. H-5′ and H-4′ showed correlations with H-2′. A full assignment of ^1^H NMR and ^13^C NMR signals was made using HMBC, HSQC, COSY, and NOESY correlations (Appendix A). The monosaccharide was glucose in chair conformation. The structure of compound **1** was established as a secoiridoid glucoside, sweroside [33]. This compound has previously been characterized in other genus of *Rubiaceae* family, but this is the first report in *Mitragyna* genus, which can allow for its best contribution in the chemotaxonomy, as it has been reported that sweroside is a precursor of indole and oxindole alkaloids [34]. The structure of the compound is presented in Figure 3.

Compound **2** (quinovic acid-3*β*-*O*-*β*-d-glucopyranoside) was isolated as a white amorphous powder. FAB-HR (*+ve*) revealed its molecular weight as 648.37 with the molecular formula C_36_H_56_O_10_. FT-IR data indicated the presence of carboxylic acid O–H (band 3275.68 cm^−1^), C–H sp3 (2942.36–2916.32 cm^−1^), C=O (1684.54 cm^−1^), and those similar to C–O (1287.57, 1252.54, 1229.50, 1211.69 cm^−1^). The molecule exhibited maximum absorption at 219 nm. However, FAB-MS (*+ve*) revealed a peak at *m/z* = 487.3225, which appeared to be quinovic acid and exhibited the loss of monosaccharide from the whole molecule. In addition, a peak at *m/z* = 603.1 that matched with monosaccharide quinovic mono acid exhibited a loss of a fragment (*m*/*z* = 44) corresponding to carboxylate. The vicinal coupling between the methyl C-29 and 30 confirmed that a triterpene is present; namely, quinovic acid. HSQC data confirmed the presence of one monosaccharide in the compound. ^13^C data indicated 36 C, 10 CH_2_, 8 quaternaries. Comparison with the reported data showed that the sugar is a d-glucose. The HMBC correlation of Glc H-1′ *δ* = 4.30 (d, *J* = 7.8 Hz) with the aglycone C-3 *δ* = 90.7 established C-1′ of the monosaccharide to C-3 at *β* position. The assignment of ^1^H NMR and ^13^C NMR signals was made using HMBC, HSQC, COSY, and NOESY correlations (Appendix A). The data of compound **2** matched with the reported quinovic acid-3*β*-*O*-*β*-d-glucopyranoside [35,36]. This compound has previously been isolated from the bark, but not from the roots, of *M. inermis* [37]. The structure of this compound is presented in Figure 4.

Compound **3** (quinovic acid-3-*O*-*β*-d-6-deoxy-glucopyranoside, 28-*O*-*β*-d-glucopyranosyl ester) was isolated as a white amorphous powder. The molecular weight and molecular formula were 794.3114 and C_42_H_51_O_15_. The FT-IR data showed the presence of carboxylic acid O–H (band at 3312.44 cm^−1^), C–H (2942.57, 2831.05 cm^−1^), and possible C–O (1113.10, 1022.79 cm^−1^). The molecule presented maximum absorption at 230 nm. The fragment ion at *m/z* = 587.3 matched to quinovic mono acid 6-deoxy-monosaccharide. The downfield chemical shift of C-6′ *δ* = 18.2, H-6′ *δ* = 1.2, d (*J* = 3.08 Hz) suggested the monosaccharide was 6′-deoxy, as 6-deoxy-d-glucose. As such, the peak at *m/z* = 587.3, which matched with quinovic mono acid 6-deoxy-monosaccharide, exhibited a loss of fragment acyl–glucoside (*m/z* = 207). The vicinal coupling between the methyl C-29 and C-30 confirmed that the triterpene is quinovic acid. The peak at *m/z* = 164.2 appears to be the fragment of 6-deoxy-d-glucopyranose. HMBC data showed correlations between H-1′ *δ* = 4.27 (d, *J* = 7.7 Hz), C-1′ *δ* = 106.6 of 6-deoxy-d-glucopyranose, and H-3 *δ* = 3.09 (d, *J* = 5.2 Hz), C-1 *δ* = 91.0 of the quinovic acid. This established the monosaccharide at *β* configuration to C-3 of quinovic ring. Furthermore, the HMBC correlations of Glucose H-1″ *δ* = 5.38 (d, *J* = 8.0 Hz), C-1″ *δ* = 95.7 with the quinovic acid C-28 *δ* = 178.37 established the monosaccharide to C-28 at *β* configuration. HSQC data confirmed the presence of two monosaccharides. Thus, instead of the FAB-MS proposal, a plausible formula for compound **3** might match C_42_H_66_O_14_. The assignment of ^1^H NMR and ^13^C NMR signals was made using HMBC, HSQC, COSY, and NOESY correlations (Appendix A). ^1^H NMR, HSQC, and HMBC 2D NMR data matched with the reported data of quinovic acid-3-*O*-*β*-d-6-deoxy-glucopyranoside, 28-*O*-*β*-d-glucopyranosyl ester [38]. This compound has been reported in *Mitragyna rotundifolia*, but this is the first report in *M. inermis* [39]. The structure of the compound is presented in Figure 4.

Compound **4** (quinovic acid 3-*O*-*α*-l-rhamnopyranosyl-(4→1)-*β*-d-glucopyranoside) was isolated as a white amorphous powder. FAB-MS (*-ve*) showed a molecular weight 794.4 and its formula was determined as C_42_H_66_O_14_. HSQC data revealed the presence of two monosaccharides. The peak at *m/z* = 749 revealed the loss of carboxylate *m/z* = 44. The peak at *m/z* = 469.22 matched to the aglycone fragment and indicated the loss of both monosaccharides; subsequently, the peak at *m/z* = 423.3 revealed the loss of carboxylate *m/z* = 44. The chemical shift of *δ*C-12 = 129.42 and *δ*C-13 = 132.47 are typical of the C=C of quinovic acid. The vicinal coupling between the methyl C-29 and 30 confirmed that the aglycone is quinovic acid. The downfield chemical shift of *δ*C-6′ = 18.4, *δ*H-6′ = 1.26 (d, *J* = 6.2 Hz) suggested that one monosaccharide is rhamnose. The HMBC correlations of one monosaccharide *δ*H-1′ = 4.69, *δ*C-1′ = 104.1 with the quinovic acid *δ*H-3 = 3.05 (dd, *J* = 11.5; 4.7 Hz), *δ*C-3 = 90.6 established the monosaccharide to C-3 at α configuration. Furthermore, the remaining monosaccharide *δ*H-1′′ = 4.57 (d, *J* = 7.8 Hz), *δ*C-1′′ = 105.8 with the first monosaccharide *δ*H-4′ = 3.59 (q, *J* = 9.3 Hz), *δ*C-4′ = 83.7 established the second monosaccharide to C-3′ at *β* configuration. HMBC showed a strong correlation with *δ*C = 72.63 and *δ*C= 68.5. The FT-IR data confirmed the presence of carboxylic acid O–H (3489.71–3316.67 cm^−1^), C–H sp3 (2948.99 cm^−1^), and C=O (1635.01 cm^−1^). This molecule presented maximum absorption at 220 nm and 230 nm. The assignment of ^1^H NMR and ^13^C NMR signals was made using HMBC, HSQC, COSY, and NOESY correlations (Appendix A). Compound **4** matched quinovic acid 3-*O*-*α*-l-rhamnopyranosyl-(4→1)-*β*-d-glucopyranoside [38]. This compound has previously been reported in *M. inermis* [40]. The structure of the compound is presented in Figure 4.

However, to understand the antiglycation activities shared by the *M. inermis* root fraction better, the isolated compounds secoiridoid glucoside sweroside (**1**), quinovic acid-3*β*-*O*-*β*-d-glucopyranoside (**2**), quinovic acid-3-*O*-*β*-d-6-deoxy-glucopyranoside, 28-*O*-*β*-d-glucopyranosyl ester (**3**), and quinovic acid 3-*O*-*α*-l-rhamnopyranosyl-(4→1)-*β*-d-glucopyranoside (**4**) were submitted for antiglycation assay. In contrast to the fraction antiglycation abilities, all of these compounds were found to be inactive against fructose-mediated BSA glycation. Therefore, these compounds might not be responsible of the antiglycation activities exhibited by *M. inermis* root fraction, which may have been due to the synergistic action of the compounds. This is the first report of non-flavonoid constituents in *M. inermis*. Further studies are needed to isolate active constituents from the potent fractions of this plant.

## 3. Materials and Methods

### 3.1. Plant Materials

*T. indica* (Voucher specimen UNB 938) leaf, stem bark, and root samples were collected from two areas at two different times. *M. inermis* (Voucher specimen UNB 939) leaf, stem bark, and root samples were collected from different areas in two periods, according to previous studies [25]. All voucher specimens were deposited at the Herbarium of Nazi Boni University.

### 3.2. Extraction and Isolation

*M. inermis* and *T. indica*, roots, stems, and leaves were extracted with 10-fold their weight in volume of acetone 70% (*v/v*) for 90 min under stirring at 1500 rpm at 37 °C. After this, the extracts were pressure-filtered, followed by centrifugation at 3800 rpm for 35 min at 4 °C. Finally, the collected supernatants were concentrated in rotavapor at 45 °C. The extracts obtained were stored at 4 °C [41].

Aqueous decoction was obtained with 10-fold the mass in distilled water under boiling for 30 min. Then, the pH of the extracts was corrected to 3–4 before liquid–liquid fractionation at equal volume successively using hexane, dichloromethane (DCM), ethyl acetate and, finally, butanol [42]. This afforded the fraction in hexane, the fraction in DCM, the fraction in ethyl acetate and, finally, the fraction in butanol, respectively. The aqueous decoction and the butanol fractions were freeze-dried, and the other fractions were concentrated in a rotavapor at 45 °C. The dry extracts and fractions were then stored at 4 °C until further use.

Extracts and fractions were dissolved in methanol and applied to a silica F254 plate. Elution was carried out using the system ethyl acetate:acetic acid:methanol:water (10:1.6:0.6:1.5). After elution over a distance of 8 cm, the spots were observed at UV 254 and 365 nm, followed by spraying with aluminum chloride and observation of the spots at 365 nm. For the same plant, extracts or fractions with similar spots or bands were combined. Hexane and dichloromethane fractions were excluded, due to a lack of remarkable flavonoid spots. All extracts were submitted to anti-glycation and cytotoxicity assays.

*M. inermis* roots combined fraction of ethyl acetate and acetone (20 g), exhibiting a relatively good anti-glycation activity, was used for the isolation of active compounds. The extract was set on the top of column with flash silica gel. Elution was done with DCM:Hexane at 50%, 75%, and 100%, then ethyl acetate:DCM at 25%, 50%, 75%, and 100%. Ethyl acetate:DCM (25:75, *v/v*) afforded compound **2** (40 mg). Subsequently, the ethyl acetate:DCM (75:25, *v/v*) afforded two sub-fractions (R152 and R157).

Subfraction R157 was separated on recycling preparative HPLC, LC-908, column ODS-H80 (C-18, 20 mm internal diameter, 250 mm length, 4 µm particle size, 80A° pore size) with a detector UV254, 0.1 abs (50 sensitivity, 3.0 mL/min, injection 10 mg/mL) using H_2_O–ACN (80:20, *v/v*), which afforded compound **4** (7 mg) in 2.6 min.

Then, sub-fraction R152 was separated on prep TLC (TLC Silica gel 60 F254, aluminum sheets 20 × 20 cm) with ACN:water (3.6:0.4) into two new sub-fractions (yellow and red). The yellow fraction on the column flash silica gel with DCM:MeOH (9:1) afforded compounds **1** (7 mg) and **3** (10 mg).

Chemical shifts (*δ*) are given in ppm relative to the residual CD_3_OD signal, and the coupling constants (*J*) are given in Hz.

### 3.3. Spectroscopic Data of the Isolated Compounds

*Secoiridoid glucoside sweroside* (**1**): Yellow, amorphous powder: FAB-HR (*+ve*): *m/z* = 359.1511[M + H]^+^; [M(C_16_H_21_O_9_) + H] = 359.1511. ^1^H NMR (400 MHz, CD_3_OD): *δ* **=** 4.67 (d, *J* = 7.9 Hz, H-1′), 3.19 (dd, *J* = 1.3, 7.9 Hz, H-2′), 3.38 (d, *J* = 8.7 Hz, H-3′), 3.29 (d, *J* = 1.2 Hz, H-4′), 3.33 (s, H-5′), 3.90 (dd, *J* = 11.9, 2.1 Hz, H-6), 3.68 (dd, *J* = 12.0, 5.7 Hz, H-6b), 5.54 (dd, *J* = 1.7, 9.9 Hz, H-1), 7.59 (d, *J* = 2.5 Hz, H-3), 3.17 (t, *J* = 2.6 Hz, H-5), 1.80 (dd, *J* = 2.6, 4.3 Hz, H-6a), 1.71 (dd, *J* = 12.8, 4.3 Hz, H-6b), 4.49 (td, *J* = 11.5, 2.8 Hz, H-7a), 4.34 (ddd, *J* = 11.1, 4.3, 2.2 Hz, H-7b), 5.59 (d, *J* = 10 Hz, H-8), 2.69 (ddd, *J* = 9.6, 5.5, 1.8 Hz, H-9), 5.33 (d, *J* = 1.9 Hz, H-10a), 5.28 (dd, *J* = 1.9, 10.3 Hz, H-10b).- ^13^C NMR (101 MHz, CD_3_OD): *δ* = 99.7 (C-1′), 74.7 (C-2′), 77.9 (C-3′), 71.5 (C-4′), 78.3 (C-5′), 62.6 (C-6′), 98.0 (C-1), 168.6 (2-C=O), 153.9 (C-3), 106.0 (4-C=), 28.4 (C-5), 25.9 (C-6), 69.7 (C-7), 133.3 (C-8), 43.8 (C-9), 120.8 (C-10). IR (KBr): V_max_ = 3326.30, 2943.41, 2831.56, 1448.66, 1115.60 cm^−1^. UV/UV–Visible (MeOH) A = 0.551 (220 nm), 0.736 (243 nm) (Appendix A).

*Quinovic acid-3β-O-β-d-glucopyranoside* (**2**): White, amorphous powder: FAB (*+ve*): *m/z* = 649.37[M + H]^+^; [M(C_36_H_56_O_10_) + H] = 649.37. ^1^H NMR (500 MHz, CD_3_OD): *δ* = 4.30 (d, *J* = 7.8 Hz, H-1′), 3.18 (dd, *J* = 8.9, 7.7 Hz, H-2′), 3.32 (dd, *J* = 1.4, 8.6 Hz, H-3′), 3.27 (dd, *J* = 8.4, 9.4 Hz, H-4′), 3.25 (dd, *J* = 5.4, 2.2 Hz, H-5′), 3.66 (dd, *J* = 11.9, 2.2 Hz, H-6a); 3.84 (dd, *J* = 11.8, 5.3 Hz, H-6b), 5.59 (dd, *J* = 5.2, 2.5 Hz, H-12), 3.15 (dd, *J* = 4.8, 12.0 Hz, H-3), 0.90 (d, *J* = 5.1 Hz, H-29), 0.92 (d, *J* = 4.8 Hz, H-30).- ^13^C NMR (151 MHz, CD_3_OD): *δ* = 106.6 (C-1′), 75.6 (C-2′), 78.2 (C-3′), 71.6 (C-4′), 77.6 (C-5′), 62.7 (C-6′), 40.38 (C-1), 19.26 (C-2), 90.7 (C-3), 31.21 (C-4), 56.90 (C-5), 18.14 (C-6), 37.63 (C-7), 40.09 (C-8), 48.01 (C-9), 38.03 (C-10), 25.75 (C-11), 130.4 (C-12), 133.8 (13-C=), 57.30 (C-14), 27.07 (C-15), 26.46 (C-16), 49.54 (C-17), 55.53 (C-18), 38.31 (C-19), 39.91 (C-20), 28.49 (C-21), 37.83 (C-22), 23.83 (C-23), 23.83 (C-24), 16.89 (C-25), 19.1 (C-26), 179.1 (27-C=O), 181.8 (28-C=O), 17.06 (C-29), 21.5 (C-30). IR (KBr): V_max_ = 3275.68, 2942.36, 2916.32, 1684.54, 1450.38, 1350.12, 1315.77, 1287.57, 1252.54, 1229.50, 1211.69, 1143.25, 1109.55, 1073.87, 980.27, 945.58, 919.39, 891.53, 770.66, 744.11, 697.76, 671.57, 653.74 cm^−1^. UV/UV–Visible (MeOH) A = 0.547 (219.0 nm), 0.472 (229.0 nm) (Appendix A).

*Quinovic acid-3-O-β-d-6-deoxy-glucopyranoside, 28-O-β-d-glucopyranosyl ester* (**3**): White, amorphous powder: FAB-HR (*+ve*): *m/z* = 795.3114[M + H]^+^; [M(C_42_H_51_O_15_) + H]= 795.3114. ^1^H NMR (400 MHz, CD_3_OD): *δ* = 4.27 (d, *J* = 7.7 Hz, H-1′), 3.17 (dd, *J* = 6.3, 9.3 Hz, H-2′), 3.38 (d, *J* = 9.1 Hz, H-3′), 2.97 (t, *J* = 9.1 Hz, H-4′), 3.25 (dd, *J* = 2.6, 9.0 Hz, H-5′), 1.26 (d, *J* = 6.1 Hz, H-6′), 5.38 (d, *J* = 8.0 Hz, H-1”), 4.04 (H-2”), 3.25 (dd, *J* = 2.6, 9.0 Hz, H-3”), 3.35 (H-4”), 3.32 (H-5”), 3.80 (d, *J* = 10.7Hz, H-6a”), 3.90 (d, *J* = 11.9 Hz, H-6b”), 5.56 (H-12), 3.09 (d, *J* = 5.9 Hz, H-3), 0.88 (s, H-29), 0.93 (d, *J* = 5.1 Hz, H-30).- ^13^C NMR (201 MHz, CD_3_OD): *δ* = 106.5 (C-1′), 75.9 (C-2′), 78.4 (C-3′), 77.1 (C-4′), 73.0 (C-5′), 18.8 (C-6′), 95.7 (C-1”), 75.2 (C-2”), 78.2 (C-3”), 71.4 (C-4”), 78.8 (C-5”), 62.7 (C-6”), 28.48 (C-1), 39.83 (C-2), 91.0 (C-3), 40.31 (C-4), 55.44 (C-5), 18.21 (C-6), 37.13 (C-7), 42.13 (C-8), 46.48 (C-9), 38.02 (C-10), 23.91 (C-11), 130.2 (C-12), 133.8 (13-C=), 56.77 (C-14), 31.16 (C-15), 30.71 (C-16), 47.91 (C-17), 51.72 (C-18), 38.15 (C-19), 40.11 (C-20), 32.10 (C-21), 37.83 (C-22), 27.13 (C-23), 27.13 (C-24), 17.07 (C-25), 19.28 (C-26), 169.5 (27-C=O), 178.3 (28-C=O), 16.95 (C-29), 21.9 (C-30). IR (KBr): V_max_ = 3312.44, 2942.57, 2831.05, 1448.12, 1113.10, 1022.79 cm^−1^. UV/UV–Visible (MeOH) A = 0.987 (220 nm), 1.058 (230 nm) (Appendix A).

*Quinovic acid 3-O-α-l-rhamnopyranosyl-(4→1)-β-d-glucopyranoside* (**4**): White, amorphous powder: Negative FAB-MS *m/z* = 793.4 [M − H], [M(C_42_H_66_O_14_)-H] = 793.4, 749 [M − H − 44], molecular weight 794.4 [M], 469.22 [M − 325]. ^1^H NMR (500 MHz, CD_3_OD): *δ* = 4.69 (d, *J* = 4.3 Hz, H-1′), 3.84 (dd, *J* = 9.0, 4.6 Hz, H-2′), 3.29 (H-3′), 3.60 (q, *J* =9.3 Hz, H-4′), 3.75 (dd, *J* = 3.4, 6.1 Hz, H-5′), 1.29 (d, *J* = 6.2 Hz, H-6′), 4.57 (d, *J* = 7.8 Hz, H-1″), 3.20 (dd, *J* = 7.9, 8.4 Hz, H-2″), 3.35 (dd, *J* = 8.7, 9.8 Hz, H-3″), 3.82 (dd, *J* = 9.0, 4.6 Hz, H-4″), 3.25 (dd, *J* = 5.6, 2.9 Hz, H-5″), 3.87 (dd, *J* = 11.6, 5.5 Hz, H-6a″), 3.68 (dd, *J* = 5.2, 3.2 Hz, H-6b″), 5.58 (d, *J* = 5.0 Hz, H-12), 3.05 (dd, *J* = 11.5; 4.7, H-3), 0.89 (d, *J* = 4.1 Hz, H-29), 0.91 (s, 30).- ^13^C NMR (201 MHz, CD_3_OD): *δ* = 104.16 (C-1′), 72.55 (C-2′), 71.46 (C-3′), 83.56 (C-4′), 68.56 (C-5′), 18.35 (C-6′), 105.68 (C-1″), 76.07 (C-2″), 78.14 (C-3″), 72.31 (C-4″), 78.02 (C-5″), 62.69 (C-6″), 39.73 (C-1), 24.95 (C-2), 90.4 (C-3), 40.16 (C-4), 56.59 (C-5), 17.99 (C-6), 37.81 (C-7), 40.45 (C-8), 40.62 (C-9), 38.02 (C-10), 23.85 (C-11), 129.9 (C-12), 132.5 (13-C=), 69.18 (C-14), 26.62 (C-15), 25.91 (C-16), 47.98 (C-17), 55.77 (C-18), 38.41 (C-19), 39.96 (C-20), 31.61 (C-21), 37.87 (C-22), 24.01 (C-23), 24.01 (C-24), 17.03 (C-25), 18.26 (C-26), 179.50 (27-C=O), 179.50 (28-C=O) 16.89 (C-29), 21.58 (C-30). IR (KBr): V_max_ = 3489.71, 3316.67, 2948.99, 1635.01, 1487.88, 1446.00, 1410.99, 1124.27, 1104.51, 1014.47, 914.64, 899.91, 886.09, 871.89, 814.43, 802.01, 704.30, 692.26, 682.33, 655.68 cm^−1^. UV/UV–Visible (MeOH) A = 0.809 (220 nm), 0.803 (230 nm) (Appendix A).

### 3.4. Anti-Glycation Activity Assay

The in vitro anti-glycation activity of both *M. inermis* and *T. indica* fractions, as well as compounds isolated from *M. inermis* roots, was determined according to a previously adopted method [43], with minor modifications. Initially, the fractions (**Frac1**–**8**) and isolated compounds (**1**–**4**) were dissolved in DMSO and tested at different concentrations (250, 500, and 1000 µg/mL; 0.03–1 mM, respectively) using a BSA–fructose glycation model. The flavonoid rutin and BSA incubated in sodium phosphate buffer (0.1 M) were used as positive and negative controls, respectively. Briefly, 10 mg/mL bovine serum albumin (BSA; Sigma Aldrich, St. Louis, Missouri, USA), 500 mM D-fructose (Merck, Darmstadt, Germany), and the fractions were incubated in a 96-well black fluorescent plate for 7 days at 37 °C in a dark sterile environment. Sodium phosphate buffer (0.1 M, pH 7.4), containing sodium azide (0.1 mM) to prevent microbial growth, was used as a medium. Later, the fluorescence of the BSA–fructose glycation was measured using a Spectra Max Spectrofluorometer (Applied Biosystems, Waltham, MA, USA) at 330/440 nm excitation and emission wavelengths (Appendix A). The percent inhibition was calculated using the following formula:$$\%\;\mathrm{Inhibition}\;=\left(1-\frac{\mathrm{Fluorescence}\;\mathrm{intensity}\;\mathrm{of}\;\mathrm{the}\;\mathrm{solution}\;\mathrm{with}\;\mathrm{treatment}\;}{\mathrm{Fluorescence}\;\mathrm{intensity}\;\mathrm{of}\;\mathrm{the}\;\mathrm{control}\;\mathrm{solution}\;}\;\right)\times 100$$

### 3.5. Cytotoxicity Assay

#### 3.5.1. Cell Culture Procedure

Mouse fibroblasts (NIH-3T3, ATCC^®^ CRL-1658™) and human hepatoma (HepG2, ATCC^®^ HB-8065™) cells were cultured and maintained according to a previously described method [30,43]. Briefly, normal fibroblasts and hepatocytes were cultured in 75 cm^2^ cell culture flasks in DMEM and MEM (Gibco, Thermo-Fischer Scientific, Waltham, MA, USA), respectively, supplemented with 10% fetal bovine serum (FBS, PAA Laboratory GmbH, Colbe, Germany). The whole cell culture procedure was performed under sterile cell culture conditions. Then, incubation was carried out in a 5% CO_2_ humified atmosphere at 37 °C. Before cell harvesting, both fibroblasts and hepatocytes were cultured to 80% confluency, and their morphological assessment was carried out using an inverted light microscope (Nikon E2000, Minato City, Tokyo, Japan).

#### 3.5.2. Cytotoxicity Procedure

Cellular toxicity of both *M. inermis* and *T. indica* fractions was tested by an MTT-based metabolic assay against mouse fibroblasts (NIH-3T3) and human hepatocellular carcinoma (HepG2) cell lines. Briefly, NIH-3T3 (10 × 10^4^ cells/mL), and HepG2 (7 × 10^4^ cells/ mL) were seeded in 96-well sterile cell culture plates and cultured overnight. The cells were treated with various fractions (**Frac1**–**8**) at different concentrations (15.6, 31.25, 62.5, 125, 250, 500, and 1000 µg/mL) for 24 h. After incubation, media were replaced with 100 µL MTT (0.5 mg/mL) in each well, and incubated for 3–4 h in a 5% CO_2_ atmosphere at 37 °C. The formazan crystals produced by succinate dehydrogenase activity (a mitochondrial enzyme) in viable cells were dissolved by adding DMSO (100 µL) in each MTT-treated well. Then, the viability of cells under different treatments was determined by taking the absorbance at 540 nm using a microplate reader (Varioskan LUX multimode microplate reader, Thermo Fisher Scientific, USA). Untreated cells were considered as controls. The toxicity effect of plant fractions on the viability of 3T3 and HepG2 cells was calculated by using the following formula:$$\%\;\mathrm{Inhibition}=100-\frac{\;\left(\mathrm{Absorbance}\;\mathrm{of}\;\mathrm{test}\;\mathrm{fraction}\;–\mathrm{Absorbance}\;\mathrm{of}\;\mathrm{blank}\right)}{\mathrm{Absorbance}\;\mathrm{of}\;\mathrm{control}\;–\mathrm{Absorbance}\;\mathrm{of}\;\mathrm{blank}}\times 100$$

### 3.6. Statistical Analysis

The GraphPad Prism 8 Statistical Software Package was used for all analyses and graph plotting. The statistical analyses were carried out by one-way ANOVA followed by Tukey’s post hoc test for antiglycation. Paired *t*-test were carried out for cytotoxicity by comparison of all fractions to **Frac 1** Values are presented as Mean ± SD. The level of significance was set as *p* < 0.05.

## 4. Conclusions

Plants are an important source of medicinally active compounds, which can help in treating a number of diseases, including diabetes. Therefore, in the current study, various fractions from different parts (i.e., leaves, stem, and roots) of *M. inermis* and *T. indica* plants were prepared, and their in vitro antiglycation potential against a BSA glycation model fructose-mediated was tested. Among the *T. indica* fractions, **Frac7** showed a significant antiglycation activity, as compared to **Frac8**, **Frac5**, and **Frac6**, derived from roots and leaves, respectively. Leaf-derived fractions **Frac2** and **Frac1** of *M. inermis* also showed significant antiglycation activity, as compared to fractions **Frac3** and **Frac4**, derived from root and stem, respectively, which showed a slightly lower inhibition against the fructose-BSA glycation. In addition, fractions were also evaluated for their cytotoxicity effect against the mouse fibroblasts (NIH-3T3) and human hepatoma (HepG2) cell lines. All fractions were found to be non-cytotoxic up to a concentration of 125 µg/mL; however, some fractions were also found to be safe up to concentrations of 250, 500, or 1000 µg/mL against both cell lines. Furthermore, we isolated compounds, including sweroside and triterpenoids, which, when tested for the inhibition of fructose-mediated BSA glycation, were found to be inactive. Further study of *M. inermis*, and *T. indica* plant fractions is still necessary, in order to identify potential candidates for the prevention and treatment of diabetes, along with its associated complications.

## Figures and Tables

**Figure 1 molecules-28-00393-f001:**
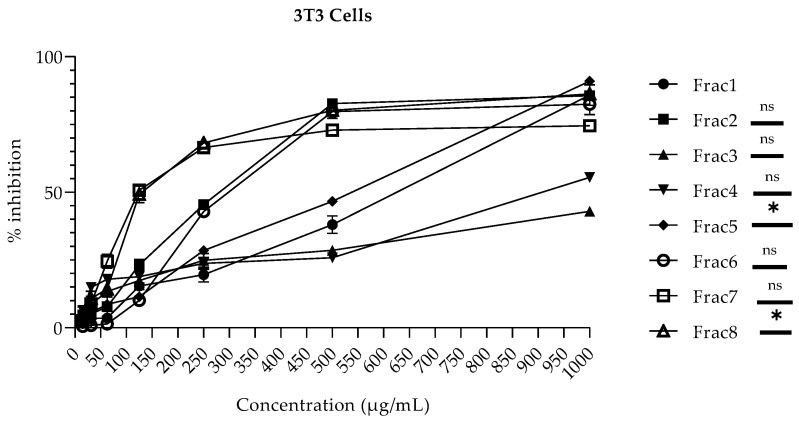
Cytotoxicity of *M. inermis* and *T. indica* fractions on 3T3 cell Line: No significant difference: ^ns^, *p* > 0.05; significant differences: *, *p <* 0.05. **Frac1**: decoction of *M. inermis* leaves; **Frac2**: additional ethyl acetate + butanol + acetone fractions of *M. inermis* leaves; **Frac3**: additional ethyl acetate + acetone fractions of *M. inermis* stem; **Frac4**: additional ethyl acetate + acetone fractions of *M. inermis* root; **Frac5**: additional ethyl acetate + butanol fractions of *T. i*ndica leaves; **Frac6**: acetone fraction of *T. indica leaves*; **Frac7**: additional ethyl acetate + acetone fractions of *T. i*ndica stem; **Frac8**: additional ethyl acetate + acetone fractions of *T. indica* root; *M. inermis*, *Mitragyna inermis*; *T. indica*, *Tamarindus indica*.

**Figure 2 molecules-28-00393-f002:**
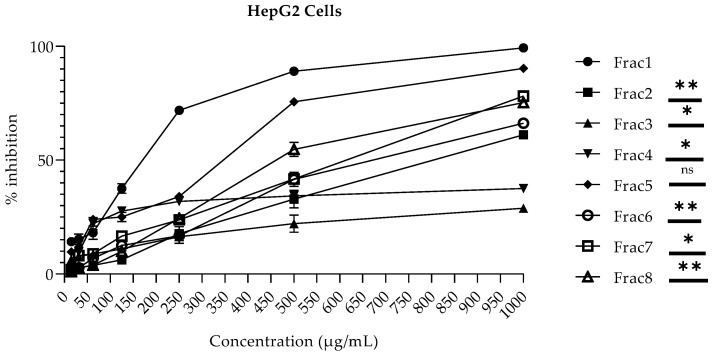
Cytotoxicity of *M. inermis* and *T. indica* fractions against HepG2 cell lines: No significant difference: ^ns^, *p >* 0.05; significant differences: *, *p <* 0.05; **, *p <* 0.01. **Frac1**: decoction of *M. inermis* leaves; **Frac2**: additional ethyl acetate + butanol + acetone fractions of *M. inermis* leaves; **Frac3**: additional ethyl acetate + acetone fractions of *M. inermis* stem; **Frac4**: additional ethyl acetate + acetone fractions of *M. inermis* root; **Frac5**: additional ethyl acetate + butanol fractions of *T. i*ndica leaves; **Frac6**: acetone fraction of *T. indica leaves*; **Frac7**: additional ethyl acetate + acetone fractions of *T. i*ndica stem; **Frac8**: additional ethyl acetate + acetone fractions of *T. indica* root; *M. inermis*, *Mitragyna inermis*; *T. indica*, *Tamarindus indica*.

**Figure 3 molecules-28-00393-f003:**
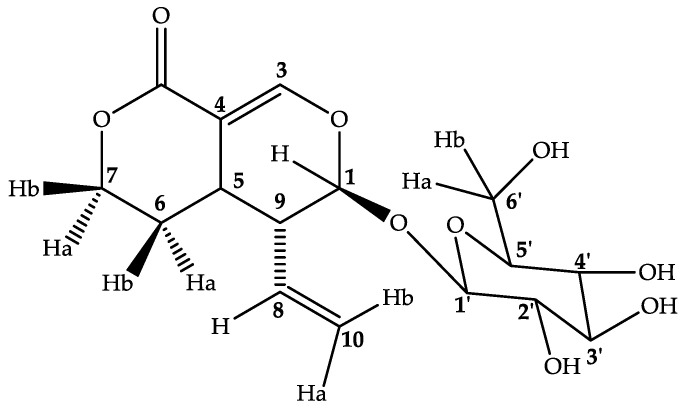
Structure of compound **1**.

**Figure 4 molecules-28-00393-f004:**
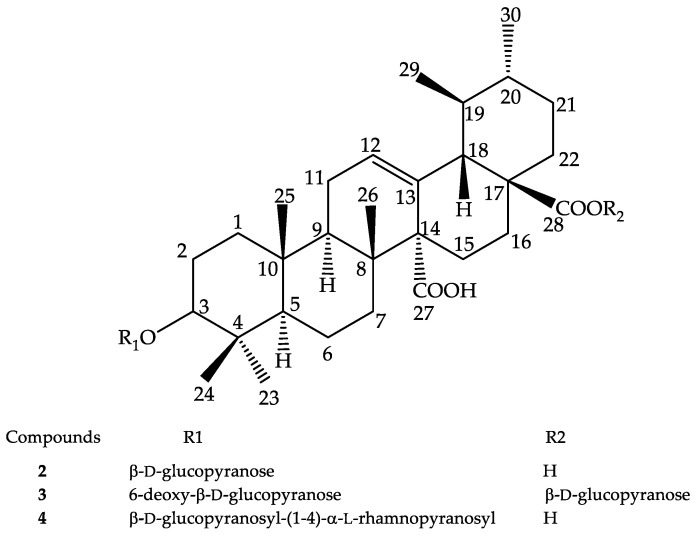
The triterpenoid glycosides obtained from *M. inermis* roots.

**Table 1 molecules-28-00393-t001:** In vitro anti-glycation activity of *M. inermis* and *T. indica* plant fractions.

Fraction (*M. inermis* or *T. indica*)	Concentration (µg/mL)	% Inhibition ±SD
**Frac1**	1000	82.87 ± 0.63 *
	500	77.94 ± 0.66 *
	250	71.86 ± 1.01 *
**Frac2**	1000	85.50 ± 0.17 ***
	500	84.30 ± 0.20 ***
	250	82.63 ± 0.10 ***
**Frac3**	1000	68.04 ± 0.78 ^ns^
	500	67.64 ± 1.09 ^ns^
	250	61.95 ± 2.07 ^ns^
**Frac4**	1000	68.34 ± 0.47
	500	64.74 ± 0.29
	250	53.99 ± 0.58
**Frac5**	1000	84.97 ± 0.26 **
	500	81.59 ± 0.75 **
	250	74.77 ± 0.80 **
**Frac6**	1000	89.66 ± 0.23 ***
	500	87.72 ± 0.55 ***
	250	81.19 ± 1.49 ***
**Frac7**	1000	91.82 ± 0.21 ****
	500	90.60 ± 0.28 ****
	250	88.12 ± 0.23 ****
**Frac8**	1000	89.03 ± 0.25 ***
	500	87.56 ± 0.001 ***
	250	85.45 ± 0.46 ***

Inhibitory effect of all fractions was compared to the % inhibition of **Frac4**. No significant difference: ^ns^, *p* ˃ 0.05; significant differences: *, *p* ˂ 0.05; **, *p* ˂ 0.01; ***, *p* ˂ 0.001; ****, *p* ˂ 0.0001. **Frac1**: decoction of *M. inermis* leaves; **Frac2**: additional ethyl acetate + butanol + acetone fractions of *M. inermis* leaves; **Frac3**: additional ethyl acetate + acetone fractions of *M. inermis* stem; **Frac4**: additional ethyl acetate + acetone fractions of *M. inermis* root; **Frac5**: additional ethyl acetate + butanol fractions of *T. i*ndica leaves; **Frac6:** acetone fraction of *T. indica leaves*; **Frac7**: additional ethyl acetate + acetone fractions of *T. indica* stem; **Frac8**: additional ethyl acetate + acetone fractions of *T. indica* root; *M. inermis*, *Mitragyna inermis*; *T. indica*, *Tamarindus indica*.

## Data Availability

Please feel free to email the correspondence authors for any additional data.

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
