# Peer review of "Inhibition of Advanced Glycation End-Products by Tamarindus indica and Mitragyna inermis Extracts and Effects on Human Hepatocyte and Fibroblast Viability"

_molecules, 2023, doi:10.3390/molecules28010393_

Round 1

Reviewer 1 Report

The manuscript entitled ‘Secoiridoid glucoside sweroside from Mitragyna. inermis and potential AGEs inhibitory effects of Mitragyna inermis and Tamarindus indica fractions’ is a manuscript describing the potential beneficial effects of plant-derived preparations against AGEs and the influence on cellular proliferation.

In the manuscript (pdf), I made some notes and suggestions (clear blue colour) that might be useful to improve the manuscript.

Critical Points

1.      The title should be improved, a suggestion could be “Inhibition of Advanced Glycation End-Products by Tamarindus indica and Mitragyna inermis extracts and Effects on Human Hepatocyte and Fibroblast viability.”, or others.

2.      The acronyms for plant fractions are too complex and should be changed in the text and in the figures. For example, EtOAc+ButOH+Ac_M.i. ... see other articles in the literature.

3.      Unfortunately, in experiments, there is no positive control, with which to compare the antiglycant effects of plant-derived fractions; an example of positive antiglycant agents often used in experiments is aminoguanidine. In the absence of a positive control, the meaning of the experiments is very low.

4.      Results and Discussion L.114-116. Reformulate the sentence by writing that other studies are needed, "the data suggest that...”

5.      Section 2.2 L.118 The title should be improved. For example, ‘Cytotoxicity profile of Mitragyna inermis and Tamarindus indica fractions in HepG2 and NIH-118 3T3 cells’

6.      L.129-L.149. Sentences are unclear, difficult to read, and should be changed.

7.      Table 1 and Figure 1 reported the same data, put only one of these.

8.      Figures 2 and 3. The x axis is expressed in logarithm? Is there a significant difference between inhibitions caused by different concentrations?

9.      L. 258 - L. 268. In what range of concentrations were isolated compounds?

10.  The abbreviation of minute is min using the SI system. Please, check the test.

11.  L. 346-L349. How much was the fluorescence of the fructose solutions?

12.  There are various typographical errors and, in general, several sentences are not very clear and should be improved.

Reviewer 2 Report

To mention in the Abstract section that the new isolated compounds are not useful does not make any sense.

The chemical nomenclature should be revised.

The structures of compounds 1 and 2 are not clear described. Figure 4 is not representative.

Table 2 and 3 does not contain all NMR information.  

Figure 5 does not have connection with the main text.

Where are the structures of the compounds 3 and 4?

The spectral data are inappropriately presented and uncompleted.

Round 2

Reviewer 1 Report

The manuscript entitled “Inhibition of Advanced Glycation End-Products by Tamarindus indica and Mitragyna inermis extracts and Effects on Human Hepatocyte and Fibroblast viability” is a manuscript relating the potential beneficial effects of plant-derived preparations against AGEs and the influence on cell proliferation.

This is the second round of revision. Effectively, the authors improved the manuscript; however, some important critical points remain. The general points that should be improved are underlined in clear blue in the revised version of the attached manuscript (R2).

Critical Points

1.      Abstract. The first sentences are unclear (L. 18-L.20); the term “fractions” should be explained in the text.

2.      L. 88 – L. 91. Fractions must also be identified in the text of the manuscript when reported for the first time in the text; what is Fra1 …, Frac2, ... etc.?

3.      The sense of sentences reported in L. 101 – L. 104 is unclear. Moreover, what is the meaning of “whole fraction”? L. 109

4.      L. 120 - L. 122 The sentence should be removed or at least, improved. The same is true for the sentence in L. 126 - L. 127.

5.      Figures 1 and Figure 2. The x-axis is not reported in linear arithmetic format because the range from 500 - 100 cannot be the same as that from 31,25 and 62,5 … This is not clear.

6.      L. 385 – L. 388. From the text: “The percentage of inhibition was calculated using the following formula: % Inhibition = (1−Test fluorescence fructose fluorescence −Control fluorescence) x 100”; This formula is not correct … What is control fluorescence? The formula should be the following: % inhibition = [1 - (fluorescence intensity of the solution with treatment/fluorescence intensity of the control solution)] x 100%.

Reviewer 2 Report

The manuscript has been significantly improved in terms of the clarity of the presented information. There are still some typographical errors.

Author Response

Please, we corrected as possible typographical errors in the manuscript.